# Pediococcus Pentosaceus from the Sweet Potato Fermented Ger-Minated Brown Rice Can Inhibit Type I Hypersensitivity in RBL-2H3 Cell and BALB/c Mice Models

**DOI:** 10.3390/microorganisms9091855

**Published:** 2021-08-31

**Authors:** Kyu-Ree Dhong, Hye-Jin Park

**Affiliations:** 1Department of Life Science, College of BioNano, Gachon University, 1342 Seongnam-daero, Sujeong-gu, Seongnam-si 13120, Korea; savanna123@gachon.ac.kr; 2Department of Food Science and Biotechnology, College of BioNano Technology, Gachon University, 1342 Seongnam-daero, Sujeong-gu, Seongnam-si 13120, Korea

**Keywords:** anti-allergy, germinated brown rice fermented with lactic acid bacteria, basophil, type I hypersensitive disease, mast cell

## Abstract

In this study, the effect of GBR fermented with the *Pediococcus pentosaceus* SP024 strain on IgE/Ag mediated passive cutaneous anaphylaxis (PCA) was investigated. Protocatechuic acid and trans-ferulic acid levels in GBR-SP024 increased more than those in unfermented GBR, respec-tively. The inhibitory activity of GBR-SP024 on β-hexosaminidase release and the level of proin-flammatory cytokine mRNA expression (tumor necrosis factor-α (TNF-α) and interleukin 4 (IL-4)) was observed in IgE/Ag-stimulated RBL-2H3 cells. Western blot analysis showed that GBR-SP024 significantly inhibited the phosphorylation of the linker for activation of T cell (LAT) and nuclear factor-κB (NF-κB) in IgE/Ag-stimulated RBL-2H3 cells. Further, we investigated the anti-allergic effect of GBR-SP024 using PCA murine model. The number of infiltrated immune cells and degranulated mast cells in GBR-SP024 treated dermis was lower than that in the GBR-treated mice. In addition, mRNA expression of 5-lipoxygenase (5-LOX) in the dermis of ear tissue declined in the GBR-SP024–treated group, compared to that in the GBR group. GBR-SP024 was also more effective than GBR at reducing the levels of IL-33 protein expression in IgE/Ag-stimulated BALB/c mice. Our study suggests the potential usage of GBR-SP024 as a dietary supplement or an adjuvant for treating IgE-dependent-allergic diseases.

## 1. Introduction

Allergic asthma, allergic rhinitis, urticaria, anaphylaxis, and atopic dermatitis are known to be caused by allergic inflammation following IgE-dependent mast cell activation. Since the year 2000, the prevalence of allergic inflammatory diseases has rapidly increased. Approximately 20% or more of the US population currently suffers from allergic rhinitis and atopic dermatitis. Between 2007 and 2009, the mortality rate of asthma per 1000 persons in the US was 0.15, and more than 500 people die from anaphylaxis annually in the United States [1]. Upon IgE activation, mast cells immediately secrete preformed and newly synthesized mediators such as histamine, leukotrienes, prostaglandins, proteases, and cytokines, giving rise to acute reactions, such as vasodilation and bronchoconstriction [2]. In addition, allergic responses also trigger the activation and infiltration of various inflammatory cells, including mast cells and lymphocytes. The rapid release of mediators and cytokines by mast cells is considered to induce and prolong these responses, leading to chronic inflammation [3]. In atopic dermatitis, chronic disease is aggravated by repeated acute responses to ambient factors. Therefore, inhibition of the immediate phase reaction is the most important challenge in the treatment of allergic inflammatory diseases [4].

*Oryza sativa* L. (common name: rice) is a staple food mostly consumed in East, South, and Southeast Asia. Notably, unpolished (brown) rice is thought to have more nutritional value due to the enrichment of fiber, iron, vitamins, and minerals in the outer bran layer of the rice grain [5]. Furthermore, the consumption of germinated brown rice (GBR) with the outer bran layer intact can promote health and quality of life [6]. GBR is considered a functional food because it is easy to digest and absorb and contains many nutrients, including γ-aminobutyric acid (GABA) and ferulic acid. GBR has been reported to possess anti-inflammatory and immune regulatory properties, enhance brain functions, and diminish blood lipid levels [5,7]. During germination, the bioavailability of nutrients is enhanced by the neutralization of phytic acid. The nutrients in unsprouted grains are poorly absorbed by the body, and incompletely digested proteins can irritate the intestines, resulting in inflammation and allergic reactions. Germination of rice promotes the absorption of proteins, vitamins, and enzymes during digestion by neutralizing phytic acid [5]. Although rice bran is known to contain polyphenolic compounds with strong antioxidative activities, people with rice allergies are unable to consume whole rice. This problem can be solved by fermenting rice [8].

The process of fermenting rice with bacteria or fungi can enhance its bioactivity. Indeed, Han et al. reported that fermenting rice bran can suppress allergic reactions [8]. Additionally, rice bran fermented with *Saccharomyces cerevisiae* possesses anti-stress and anti-fatigue effects [9]. Furthermore, fermented rice bran extracts suppress melanogenesis by downregulating microphthalmia-associated transcription factor (MITF) and reducing cytotoxicity [10]. In this study, we fermented germinated *Oryza sativa* L. (GBR) with several LAB strains to enhance the nutritional value and biological activity of GBR. We investigated the anti-allergic activity of LAB-fermented germinated brown rice against type I hypersensitivity in vitro and in vivo.

## 2. Materials and Methods

### 2.1. Preparation and Extraction of GBR Fermented with Different LAB Strains

GBR and fermented GBR (*Lactobacillus paraplantarum* SC61 and *Pediococcus pentosaceus* SP024) were provided by the Cell Activation Research Institute (Seoul, Korea). To prepare GBR extracts for liquid fermentation, 20 times the volume of distilled H_2_O was added to the GBR and heated at 121 °C for 15 min. GBR was fermented with LAB (*L. paraplantarum* SC61 and *P. pentosaceus* SP024) as described in a previous study [11]. The GBR extracts were centrifuged at 13,100× *g* for 10 min to obtain the supernatant, which was then fermented with each LAB strain. Inoculated GBRs were heat-killed at 100 °C for 10 min and sonicated for 3 min (Sonics & Materials, Inc., Newtown, CT, USA).

Solid-phase fermentation of GBR with LAB (*L. paraplantarum* SC61 and *P. pentosaceus* SP024) was performed as described previously [11,12]. GBR-SC61 and GBR-SP024 (100 g) were extracted with 70% EtOH for 2 h at 50 °C. The ethanol extract was then concentrated using a rotary evaporator. 40g of the extractions were dissolved in 70% ethanol.

### 2.2. 2,2-Diphenyl-1-Picrylhydrazyl (DPPH) Radical Scavenging Activity

The free radical scavenging activity of the samples was evaluated as described in a previous study [11]. DPPH solution (100 μL, 100 μM) was added to 100 μL of 10 mg/mL fermented solution and 1 mM ascorbic acid solution, which was used as a positive control. After 30 min of incubation at 37 °C, the absorbance was measured at 517 nm using a microplate reader (Epoch, BioTek Instruments, Winooski, VT, USA). Radical scavenging activity was calculated as DPPH radical scavenging activity (%) = 100 − [(Abs sample − Abs blank) × 100]/Abs control, where Abs sample is the absorbance value of the sample plus DPPH solution, Abs blank is the absorbance value of the blank (sample plus methanol), and Abs control is the absorbance value of the control (DPPH solution plus methanol).

### 2.3. Total Phenol Content

The total phenolic content in the samples was measured by the Folin-Ciocalteu colorimetric method as described in a previous study [13]. Each sample (50 μL) was blended with 0.2 N Folin-Ciocalteu phenol reagent (100 μL) for 5 min, and then 750 μL of sodium carbonate solution was added. After 1 h incubation at room temperature, the absorbance was measured at 750 nm using a microplate reader (Epoch). Data are expressed as mg of gallic acid equivalents (GAE) per gram of GBR, GBR-SC61, and GBR-SP024.

### 2.4. HPLC

The contents of *trans*-ferulic acid and protocatechuic acid in GBR and GBR-SP024 were evaluated by HPLC according to Lee’s and Mitani’s groups [14,15]. The sample solutions were separated using a Hydrosphere C18 column (250 × 4.6 mm, 5 μm; YMC). The column temperature was set to 30 °C. The mobile phase consisted of acetonitrile (A) and water with 0.1% formic acid (B) at a flow rate of 0.7 mL/min. Gradient elution was performed as follows: from 0 to 5 min, linear gradient from 5 to 9% solvent A; from 15 to 22 min, 11% solvent A; from 22 to 38 min, linear gradient from 11 to 18% solvent A; from 38 to 88 min, linear gradient from 18 to 80% solvent A. Phenolic compounds were detected at 280 nm by UV spectroscopy.

### 2.5. Cell Culture and Viability Assay

Rat basophilic leukemia cells (RBL-2H3, catalog number KCLB no. 22256, American Type Culture Collection, Manassas, VA, USA) were grown in Eagle’s Minimum Essential Medium (EMEM) (Welgene, Daegue, Korea) supplemented with 15% heat-inactivated FBS (Welgene) and 1% penicillin streptomycin (Welgene). Cells were grown in 75 cm^2^ culture flasks at 37 °C with 5% CO_2_ in a humidified incubator (Forma 3111, Thermo Fisher Scientific, Waltham, MA, USA). Cell viability was evaluated using the Cell Counting Kit-8 assay (Dokindo Laboratories, Kumamoto, Japan) [16]. The absorbance was measured at 450 nm using a microplate reader (Epoch). Cell viability was calculated as a percentage relative to that of the untreated RBL-2H3 cells.

### 2.6. β-Hexosaminidase Release Assay

RBL-2H3 cells (1 × 10^6^ cells/well) were seeded in 24-well plates and treated with 200 ng/mL of DNP-specific IgE (Sigma-Aldrich, St. Louis, MO, USA). Cells were treated with or without GBR, GBR-SC61, GBR-SP024 (200 μg/mL), or PP2 (Calbiochem, La Jolla, CA, USA) which is an Src tyrosine kinase inhibitor. After incubation for 30 min at 37 °C, the cells were exposed to 200 ng/mL of DNP-BSA antigen (Sigma-Aldrich) for 15 min at 37 °C. The degree of degranulation was determined by measuring the amount of β-hexosaminidase released, as previously described [13]. The absorbance at 405 nm was measured using a microplate reader (Epoch).

### 2.7. Experimental Murine Model

BALB/c mice (aged 6–8 weeks) were purchased from Orient Bio (Seongnam, Korea). Mice were raised in cages at 23 ± 2 °C and 55% humidity. Six mice were used for animal experiment at each group. The animal experiments were performed under the institutional guidelines of the Institutional Animal Care and Use Committee at Gachon University.

### 2.8. Passive Cutaneous Anaphylaxis (PCA)

PCA was induced in mice as previously described [13,16]. DNP-specific IgE (0.5 μg; Sigma-Aldrich) was intradermally injected into mouse ears. After 24 h, GBR, GBR-SC61, and GBR-SP024 (200 mg/kg) were orally administered to mice 1 h before stimulation with 1 mg/mL of DNP-BSA containing 1% Evans blue dye. Each side of whole ear was cut to measure the amount of extravasated dye. To extract the dye, whole ears were immersed in 700 μL of formamide at 63 °C. Absorbance was read using a microplate reader (Epoch) at 620 nm.

### 2.9. Hematoxylin and Eosin (H&E) Staining

Formalin-fixed whole ear tissues were fixed in paraffin blocks and cut into 5 μm sections. Each section was stained with hematoxylin (Polysciences, Inc., Warminster, PA, USA) and eosin (Shimakyu Pure Chemicals, Osaka, Japan). Images were taken using a light microscope (Nikon Instruments Inc., Melville, NY, USA) to measure ear thickness. Digital micrographs were taken from representative areas at 200× fixed magnification. Ear thickness was determined using Axiovision 4.7 image analysis software (Carl Zeiss Vision GmbH, Munich, Germany).

### 2.10. Toluidine Blue Staining

Mast cells in each side of whole ear tissue was stained with toluidine blue (Sigma-Aldrich), as previously described [16]. The total number of mast cells and degranulated mast cells were counted from the images taken under a light microscope (Nikon Instruments Inc., Melville, NY, USA) at 200× magnification.

### 2.11. Immunohistochemistry

Immunohistochemical staining was performed as previously described [17]. Tissue sections of whole ear were deparaffinized and rehydrated with xylene and graded ethanol solutions. The sections were incubated with anti-IL-33 (Abcam, Cambridge, MA, USA) for 1 h in PBS and anti-NF-κB p65 (Cell Signaling Technology, Beverly, MA, USA) in PBS. After washing, the sections were incubated with anti-rabbit antibodies (Dako, Santa Clara, CA, USA) for 30 min and counterstained with Mayer’s hematoxylin. Images were taken using a light microscope at 200× (Nikon Instruments Inc.). Stained cells were counted using ImageJ.

### 2.12. Reverse Transcriptase Polymerase Chain Reaction (RT-PCR)

Total RNA was extracted from the RBL-2H3 cells and the whole ear tissue from BALB/c using TRIzol reagent (Invitrogen, Carlsbad, CA, USA) and reverse transcribed using a Revertra Ace qPCR RT kit (Toyobo Biologics Inc., Osaka, Japan) as described previously [16]. Polymerase chain reaction was performed at 94 °C for 2 min, 94 °C for 30 s, 55 °C for 30 s, and 68 °C for 1 min for 30 cycles. The following primers were used: rat TNF forward 5′-CAC CAC GCT CTT CTG TCT ACT GAA C-3′, reverse 5′-CCG GAC TCC GTG ATG TCT AAG TAC T-3′; rat IL-4 forward 5′-ACC TTG CTG TCA CCC TGT TC-3-3′, reverse 5′-TTG TGA GCG TGG ACT CAT TC-3-3′; rat GAPDH forward 5′-CTT CAC CAC CAT GGA GAA GGC TG-3-3′, reverse 5′-GAC CAC AGT CCA TGC CAT CAC TG-3-3′; mouse COX-2 forward 5′-AAC CGT GGG GAA TGT ATG AGC A-3′, reverse 5′-AAC TCT CTC CGT AGA AGA ACC TTT TCC A-3′; mouse 5-LOX forward 5′-GAC TTC CAC GTC CAT CAA A-3′, reverse 5′-GGA AAC ACA GGG AGG AAT AG-3′ (Bioneer, Daejeon, Korea). The levels of TNF-α, IL-4, COX-2, and 5-LOX mRNAs were normalized to GAPDH mRNA levels.

### 2.13. Western Blot Analysis

Protein analysis was performed as previously described [13,18]. Proteins (80 µg) from each sample were electrophoresed on 7–10% gradient polyacrylamide gels and then transferred to nitrocellulose membranes. The membranes were blocked in 5% BSA for 1 h and incubated with primary antibodies against phosphorylated Syk (Cell Signaling Technology), phosphorylated LAT (Cell Signaling Technology), phosphorylated ERK (Cell Signaling Technology), phosphorylated NF-κB (Cell Signaling Technology), phosphorylated ERK1/2 (Cell Signaling Technology), and β-actin (Cell Signaling Technology). After washing with Tris-buffered saline with Tween 20 (TBST) buffer (Bio-Rad Laboratories, Hercules, CA, USA), the membranes were incubated with horseradish peroxidase-labeled secondary antibody (Cell Signaling Technology). The bands were detected using LI-COR Odyssey (LI-COR Biosciences).

### 2.14. Statistical Analysis

The results were obtained from at least three independent experiments and shown as means± standard error of means (SEM). All data were checked for normality distribution by Shapiro-wilk test using SPSS. One-way analyses of variance (ANOVA) followed by Dunnett’s *t*-tests or Duncan’s *t*-tests were used to analyze significant differences between the treatment groups. Data were analyzed using the Statistical Package for the Social Sciences software (SPSS 12; SPSS Inc., Chicago, IL, USA).

## 3. Results

### 3.1. Screening of LAB for Fermenting Germinated Brown Rice

To select the best LAB strain for fermenting GBR, we compared the DPPH activity and total polyphenol content (TPC) of each sample. A total of 55 strains were tested for GBR fermentation (data not shown). Among them, *P. pentosaceus* SC61 and *L. paraplantarum* SC024 both exerted high DPPH radical scavenging activity (67.6% ± 3.2 and 86.7% ± 2.7, respectively) and titratable acidity (0.1%, 0.1%, respectively, data not shown, Table 1) and were chosen for fermenting GBR. After solid fermentation at 37 °C for 24 h, the TPC of GBR-SC61 and GBR-SP024 was 1.1 ± 0.0 and 1.2 ± 0.0 times higher than that of GBR, respectively (*p* < 0.05 vs. GBR, Figure 1). Ferulic acid, a phenolic compound in brown rice, is known for its anti-inflammatory activity. The ferulic acid content of GBR-SP024 was 1.2 ± 0.0 times higher than that of GBR (*p* < 0.01 and *p* < 0.001, Table 2). Our data indicated that the SP024 LAB strain was more effective at increasing the TPC and antioxidant activity of GBR than the SC61 LAB strain.

### 3.2. Fermented GBR Decreased the Release of β-Hexosaminidase in IgE/Ag-Stimulated RBL-2H3

To determine whether the anti-allergic activity of GBR was enhanced after LAB fermentation, we investigated the amount of β-hexosaminidase released in IgE/Ag-stimulated RBL-2H3 cells after sample treatment. The amount of released β-hexosaminidase is often used as a degranulation indicator, as it is discharged along with histamine upon mast cell degranulation [19]. We treated the IgE-primed RBL-2H3 cells with 200 µg/mL of GBR, GBR-SC61, and GBR-SP024 prior to Ag stimulation. RBL-2H3 cells treated with GBR-SC61 and GBR-SP024 released 5.47 ± 1.4 and 4.5 ± 0.56 times less β-hexosaminidase than IgE/Ag-treated RBL-2H3 cells (*p* < 0.001 vs. IgE/Ag-stimulated control). The inhibitory activities of GBR-SC61 and GBR-SP024 were 1.9 ± 0.6 and 1.6 ± 0.7 times higher than those of GBR, respectively (*p* < 0.001 vs. IgE/Ag-stimulated control, Figure 2A). Cytotoxicity was observed in cells treated with 200 μg/mL of GBR but not in cells treated with LAB-fermented GBR (Figure 2B).

### 3.3. GBR-SP024 Decreased TNF-α and IL-4 mRNA Expression in IgE/Ag-Stimulated RBL-2H3 Cells

Next, we assessed whether GBR-SC61 and GBR-SP024 suppressed the mRNA expression of TNF-α and IL-4 in IgE/Ag-treated RBL-2H3 cells. TNF-α and IL-4 play critical roles in activating immune cells to produce other cytokines and IgE antibodies that aggravate allergic reactions [8,20]. GBR-SC61 treatment reduced TNF-α and IL-4 mRNA expression in a dose-dependent manner (58.3% ± 3.3, 44.9% ± 5.45), compared with IgE/Ag stimulation (*p* < 0.05, *p* < 0.01 vs. IgE/Ag-treated cells) (Figure 3). Similarly, GBR-SP024 treatment significantly inhibited TNF-α and IL-4 mRNA expression in RBL-2H3 cells (60.3% ± 10.9, 65.2% ± 10.6), compared with IgE/Ag stimulation (*p* < 0.01). GBR-SP024 significantly suppressed the levels of IL-4 mRNAs in IgE/Ag-treated cells, compared with GBR-SC61. While GBR-SP024 significantly decreased the levels of TNF-α mRNA in IgE/Ag-treated cell, compared with GBR, but not with GRC-SC61 (*p* < 0.01). Therefore, we chose GBR-SP024 as the primary LAB strain for further experiment.

### 3.4. GBR-SP024 Inhibited the Activation of FcεRI and NF-kB Signaling Pathway Molecules in IgE/Ag-Stimulated RBL-2H3 Cells

When Ag is crosslinked with IgE-sensitized FcεRI in mast cells, various intracellular signaling pathways are activated, leading to the secretion of granules and production of proinflammatory cytokines [21]. In the FcεRI and nuclear factor-κB (NF-κB) signaling pathways, stimulated Src family kinases (Lyn and Syk) activate downstream molecules (LAT and ERK). To determine whether GBR-SP024 affected the activation of the FcεRI and NF-κB signaling pathways, we examined whether GBR-SP024 reduced the levels of phosphorylated Syk, LAT, ERK, and NF-κB proteins. GBR-SP024 significantly decreased the level of phosphorylated Syk, LAT, ERK and NF-κB proteins, compared to IgE/Ag-stimulated controls (Figure 4B, *p* < 0.05). In addition, GBR-SP024 significantly suppressed the level of phosphorylated LAT and NF-κB proteins, compared to GBR (*p* < 0.05). These results suggested that GBR-SP024 showed anti-allergic properties in IgE/Ag-stimulated RBL-2H3 cells by suppressing the activation of signaling molecules in the FcεRI and NF-κB pathways.

### 3.5. GBR-SP024 Reduced IgE/Ag-Mediated PCA and Inflammatory Cell Infiltration in BALB/c Mice

Mast cells and basophils are the major effector cells in allergic responses and anaphylaxis reactions. To assess the anti-allergic effect of GBR-SP024, we used a murine model for PCA, which is an IgE-dependent allergic response. Furthermore, we employed a mast cell-dependent PCA model to assess whether GBR-SP024 could repress mast cell activation in vivo. IgE was injected intradermally into the ear, and then DNP-BSA containing 1% Evans blue dye was injected into the tail. GBR-SP024 treatment (200 mg/kg) significantly suppressed extravasation of Evans blue dye in the IgE/Ag-stimulated ear (*p* < 0.01, Figure 5A,B).

To ensure that GBR and GBR-SP024 were not toxic to mice, we considered lethality rates and changes in body and spleen weights. There was no change in body weight and spleen index over 8 days of treatment, suggesting that these samples were not toxic to mice (Figure 5D,E).

We further performed histopathological examination by analyzing the H&E-stained ear tissues of IgE/Ag-stimulated mice after GBR and GBR-SP024 treatment. Notable infiltration of inflammatory cells was observed in the ear epidermis and dermis of the PCA mice (Figure 5A). The number of infiltrated inflammatory cells in GBR-SP024 was 1.17 ± 0.18 times fewer than that in the IgE/Ag-stimulated control (*p* < 0.01, Figure 5D). GBR-SP024 treatment was more effective in decreasing the number of infiltrated inflammatory cells than GBR treatment (72.0% ± 7.0 vs. 92.0% ± 8, *p* < 0.01). The ear thickness in the GBR-SP024 group was 1.4 ± 0.2 times thinner than that in the PCA group (*p* < 0.01, Figure 5C).

### 3.6. Fermented GBR Suppressed Mast Cell Degranulation in the Ears of IgE/Ag-Stimulated BALB/c Mice

Mast cells play a key role in allergic responses, secreting various mediators, including histamine, prostaglandins, leukotrienes, and cytokines [22]. We checked whether GBR-SP024 could attenuate the infiltration and mast cells in the ear tissues (Figure 6A). The number of granulated (54.2% ± 8.7) and degranulated (66.0% ± 2.0) mast cells in the ear tissues of GBR-SP024-treated mice was significantly lower than that in IgE/Ag-stimulated mice (*p* < 0.01, Figure 6B,C). Furthermore, GBR-SP024 treatment significantly decreased the number of degranulated mast cells (45.8% ± 2.5), compared to GBR treatment (*p* < 0.01).

### 3.7. GBR-SP024 Decreased COX-2 and 5-LOX mRNA Expression in the Ear Tissues of IgE/Ag-Stimulated BALB/c Mice

Allergic responses are exacerbated by pro-inflammatory mediators, including prostaglandins and leukotrienes [23,24]. COX-2 converts arachidonic acid to prostaglandin in IgE-mediated mast cells. Arachidonic acids are also converted into leukotrienes by 5-LOX in IgE-mediated mast cells [25]. To determine whether GBR-SP024 affected COX-2 and 5-LOX mRNA expression, we analyzed COX-2 and 5-LOX mRNA expression in the ear tissues of IgE/Ag-stimulated mice using RT-PCR. As shown in Figure 7, GBR-SP024 significantly inhibited the mRNA expression of 5-LOX and COX-2 (63.8% ± 3.2, 87.8% ± 2.0, *p* < 0.05). Furthermore, the mRNA levels of 5-LOX and COX-2 in IgE/Ag-stimulated mice treated with GBR-SP024 (63.8% ± 3.2 and 79.9% ± 2.2) were significantly decreased compared to those in mice treated with GBR (20.1% ± 3.6 and 42.0% ± 3.3, *p* < 0.05).

### 3.8. Fermented GBR Decreased IL-33 and NF-κB Protein Expression in IgE/Ag-Stimulated BALB/c Mice

IL-33 belongs to the IL-1 cytokine family and induces allergic responses, such as asthma, allergic rhinitis, and atopic dermatitis [26]. IL-33 protein levels are upregulated in atopic dermatitis patients and in IgE/Ag-stimulated mice [27,28]. To assess the effect of GBR-SP024 on the expression of IL-33 in the ears of IgE/Ag-stimulated mice, we performed immunohistochemical staining to determine the number of IL-33-positive cells in the dermis area. IL-33 protein levels (56.7% ± 8.8) were significantly decreased in the GBR-SP024-treated mice, compared to those in the PCA model mice (*p* < 0.01, Figure 8A,C). GBR-SP024 treatment also significantly suppressed IL-33 protein expression in IgE/Ag-stimulated mice, compared with GBR treatment (*p* < 0.01).

To assess the effect of GBR-SP024 on NF-κB expression, we assessed the number of NF-κB-positive cells using immunohistochemical staining. NF-κB is activated by IgE-mediated allergic reactions through the IL-33/ST2 signaling pathway [29]. NF-κB induces the expression of many pro-inflammatory and lipid mediator genes in activated mast cells and basophils [30]. The number of NF-κB-positive cells (42.7% ± 1.5) was significantly reduced in GBR-SP024-treated IgE/Ag-stimulated BALB/c mice (*p* < 0.01, Figure 8B,D).

## 4. Discussion

Significant efforts have been made to develop novel active agents for alleviating type I hypersensitivity reactions with no side effects. In this study, we demonstrated that GBR fermented with *P. pentosaceus* SP024 suppressed type I hypersensitive reactions.

Previously, it was reported that GBR fermented with *Lactobacillus acidophilus* exerts anti-cancer activity [31]. However, there have been no studies on the anti-allergic activity of GBR fermented with LAB. Here, we fermented GBR with various lactic acid bacterial strains to enhance the bioactivity of GBR components. GBR, unpolished germinated brown rice with an intact outer bran layer, was selected as the substrate for LAB fermentation because it is rich in nutrients, such as minerals, vitamins, GABA, and polyphenolic acids (e.g., ferulic acid), compared to ordinary brown rice [5]. The concentration of phenolic compounds, such as p-coumaric acid, protocatechuic acid, and ferulic acid, increased in brown rice after germination [32]. Notably, these compounds are known to possess antioxidant, anti-inflammatory, and anti-allergic activities.

Interestingly, the content of phenolic compounds in GBR increased after LAB fermentation [33]. We found that fermentation with *P. pentosaceus* SP024 increased protocatechuic acid and ferulic acid levels in GBR. Ferulic acid is known to possess anti-allergic activity by suppressing Th2 immune response and IgE level as well as antioxidant activity by scavenging of free radicals [34,35]. Oka et al. reported that cycloartenyl ferulate suppressed mast cell degranulation in IgE-sensitized RBL-2H3 cells [36]. In addition, γ-oryzanol, a mixture of steryl ferulates, produces anti-allergic effects by inhibiting IL-4 and IgE gene expression [37]. Moreover, protocatechuic acid inhibited NF-κB signaling and inflammatory cytokine release in an asthma model [38]. We reported that *P. pentosaceus* SP024 and *L. paraplantarum* SC61-fermented GBR demonstrated increased DPPH radical scavenging activity. The increased antioxidant activity of GBR-SP024 may be due to the increased ferulic acid and protocatechuic acid content in GBR following *P. pentosaceus* fermentation. However, other bioactive compounds might also enhance the antioxidant activity of GBR. In particular, Lee et al. reported that 3-methyl-1-butanol, which possesses antioxidant properties, was found in rice following *P. pentosaceus* fermentation [39]. Moreover, phenolic esterase, found in LAB, has been reported to increase the level of polyphenol compounds in rice brans [40,41]. GBR-SP024 holds promise for the treatment of IgE-dependent allergic reactions, as both ferulic acid and protocatechuic acid are known to possess anti-allergic activities. However, the specific anti-allergic effects of GBR-SP024 and its mechanism of action have not yet been investigated in in vitro and in vivo experimental allergy models.

Mast cells and basophils are key cells in IgE-mediated hypersensitivity reactions. Ag binding to the IgE-primed FcεRI receptor leads to the activation of immunoreceptor tyrosine-based activation motifs (ITAMs), serving as a platform for spleen tyrosine kinase (Syk) to be activated. Syk, a key intracellular molecule in early FcεRI-mediated signaling, activates Gab2 and LAT. Phosphorylated LAT then recruits SLP-76, Grb2/Sos, and phospholipase Cγ (PLCγ), which induce degranulation, cytokine production, and the activation of downstream molecules such as extracellular signal-regulated protein kinases (ERK) and c-Jun N-terminal kinases (JNK) [42,43]. Indeed, a previous study reported that phosphorylated Syk induces NF-κB and NFAT activation [44].

NF-κB is a transcription factor that regulates pro-inflammatory cytokines such as TNF-α and enzymes such as COX-2 [43]. Additionally, the NF-κB signaling pathway can induce allergic reactions such as PCA [45]. Protocatechuic acid inhibits the phosphorylation of NF-κB in IgE/Ag-stimulated RBL-2H3 cells [46,47]. Therefore, it is possible that the increased protocatechuic acid levels in GBR due to SP024 fermentation were responsible for suppressing the phosphorylation of NF-κB. In IgE-dependent responses, NF-κB signaling induces the expression of pro-inflammatory cytokines [48]. In our in vitro experiments, RBL-2H3 basophils were used to investigate the anti-allergic effects of GBR-SP024 [13,49,50]. IgE/Ag binds to FcεRI in basophils, initiating degranulation and leading to the release of histamine, β-hexosaminidase, and pro-inflammatory cytokines such as TNF-α and IL-4 [45,51]. Fan et al. reported that fermented rice bran inhibited β-hexosaminidase release and pro-inflammatory cytokine production (TNF-α and IL-4) [8]. Consistent with this result, we found that GBR-SP024 notably suppressed the release of β-hexosaminidase and the production of TNF-α and IL-4.

TNF-α and IL-4 are known to be responsible for aggravating allergic responses. Several studies have demonstrated that these cytokines are associated with pathogenic features of IgE-dependent allergic diseases, such as IgE synthesis, Th2-type cytokine production, and chemokine overproduction. TNF-α is necessary for the recruitment of inflammatory cells, induction of Th2-type cytokines, and the expression of adhesion molecules [52,53,54]. IL-4 induces the Th2 response by activating naïve T cells to differentiate into allergen-specific Th2 cells and switching B cells into IgE-producing plasma cells [55,56,57]. Furthermore, ferulic acid inhibits TNF-α and IL-4 cytokine levels in IgE-dependent allergic disease experimental models [58]. In addition, ferulic acid might suppress the reactive oxygen species (ROS)/NF-κB pathway by scavenging ROS, leading to a reduction in proinflammatory cytokine production [59]. Therefore, the inhibition of IgE/Ag-induced TNF-𝛼 and IL-4 production by GBR-SP024 could have been due to increased ferulic acid levels in GBR following SP024 fermentation.

The PCA reaction is an IgE-mediated type 1 hypersensitivity in the dermis. The most common and important features of PCA include the release of active materials, such as histamine, which induce vascular permeability and the extravasation of Evans blue dye. We found that orally administered GBR-SP024 inhibited the increase in vascular permeability and the extravasation of Evans blue dye in mice with PCA and IgE-mediated type 1 hypersensitivity in the dermis. Additionally, the number of degranulated mast cells could be quantified by excising skin tissue from the reaction site and staining with toluidine blue. Consistent with the in vitro result, GBR-SP024 decreased the number of degranulated cells in the dermis of PCA mice. Treatment with GBR-SP024 was as effective as cetirizine (CZ), an established drug for type I hypersensitivity treatment. Therefore, GBR-SP024 may suppress IgE-dependent mast cell activation or protect against the initial surge in inflammatory cell infiltration.

As COX-2 and 5-LOX were both highly expressed in IgE/Ag-stimulated mice, the downregulation of either COX-2 or 5-LOX might inhibit the development of PCA [25]. COX-2 and 5-LOX are key enzymes that convert arachidonic acid into lipid mediators such as PGE2 and LTE4 [23]. Lipid mediators increased vascular permeability and recruited immune cells during IgE-dependent anaphylaxis [24,60]. Therefore, the decreased vascular permeability and suppressed immune cell infiltration observed in the GBR-SP024 treated group might be due to the inhibition of COX-2 and 5-LOX mRNA expression.

Recent studies have suggested that IL-33 is mainly involved in the development of allergic diseases [61]. IL-33 is a member of the IL-1 cytokine family and is expressed in endothelial cells, epithelial cells, and fibroblasts. Additionally, IgE/Ag-activated mast cells also release IL-33 [28]. During allergic inflammation, IL-33 promotes the production of Th2 cytokines such as IL-4 and IL-13 in basophils and mast cells by binding to suppressor of tumorigenicity 2 (ST2) receptors. Moreover, IL-33/ST2 signaling recruits MyD88 and activates MAPK and NF-κB signaling (p50 and p65) in immune cells [27,29,62]. These signaling pathways also promote IL-4 and IL-5 cytokine expression in mast cells and basophils. Here, we demonstrated that GBR-SP024 treatment was more effective than GBR treatment at suppressing IL-33 expression in PCA mice. According to Brugiolo et al., ferulic acid inhibits IL-33 cytokine levels in IgE-dependent allergic reactions in mice [35]. These studies suggest that GBR-SP024 might inhibit IL-33 expression in mast cells and endothelial cells with IgE-dependent anaphylaxis by increasing ferulic acid levels.

NF-κB is known to play key roles in asthma and anaphylaxis. As described previously, NF-κB is activated by IL-33/ST2 signaling in IgE/Ag-stimulated mast cells and basophils. Previous studies have reported that protocatechuic acid blocks the activation of the p38 MAPK and NF-kB signaling pathways and the subsequent expression of pro-inflammatory cytokines in asthmatic allergic model mice [47,63]. Therefore, GBR-SP024 might increase protocatechuic acid levels to inhibit pro-inflammatory cytokine mRNA expression (TNF-α and IL-4) in RBL-2H3 cell IgE-mediated allergic responses by suppressing activated NF-κB protein levels. Therefore, the enhanced anti-allergic activity of GBR-SP024 was most likely the result of increased levels of ferulic acid and protocatechuic acid. In this study, we assessed the anti-allergic activity of GBR-SP024. However, further study of the anti-allergic compounds presents in GBR-SP024, including ferulic acid and protocatechuic acid, is clearly necessary.

## 5. Conclusions

In this study, we found that the levels of ferulic acid and protocatechuic acid were increased in GBR after fermentation with *P. pentosaceus* SP024. GBR-SP024 showed enhanced anti-allergic activity against IgE/Ag-mediated type I hypersensitivity reactions. Furthermore, GBR-SP024 inhibited degranulation, proinflammatory cytokine expression, and the phosphorylation of Syk, ERK, and NF-κB in the FcεRI-mediated signaling pathway in IgE/Ag-stimulated RBL-2H3 cells. Additionally, GBR-SP024 inhibited the PCA reaction and the levels of IL-33 and NF-κB in IgE/Ag-stimulated BALB/c mice. These findings suggest that GBR-SP024 is a potential therapeutic agent for the treatment of IgE-dependent type I allergic diseases.

## Figures and Tables

**Figure 1 microorganisms-09-01855-f001:**
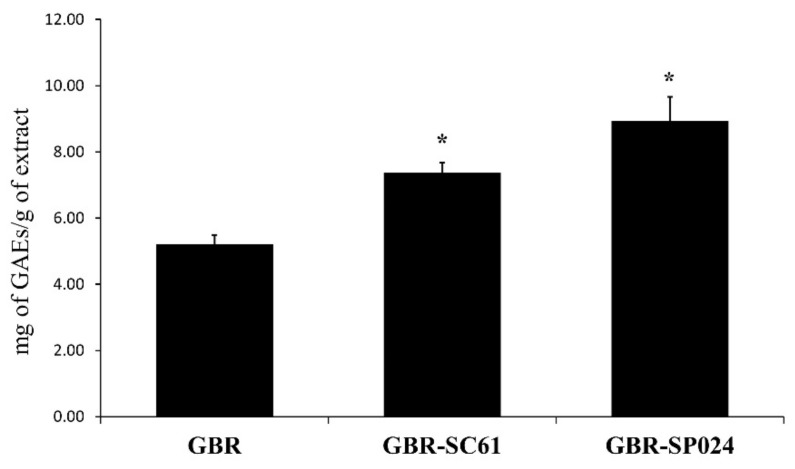
Total polyphenol content in GBR and GBR fermented with *Lactobacillus paraplantarum* SC61 and *Pediococcus pentosaceus* SP024. GBR, unfermented germinated brown rice; GBR-SC61, *L. paraplantarum* fermented GBR at 37 °C for 24 h; GBR-SP024, *P. pentosaceus* fermented GBR at 37 °C for 24 h. One-way ANOVA was used to compare group means, followed by Dunnett’s *t*-test (* *p* < 0.05 vs. GBR). Data are expressed as means ± standard error of means (SEM) from three independent experiments (*n* ≥ 3).

**Figure 2 microorganisms-09-01855-f002:**
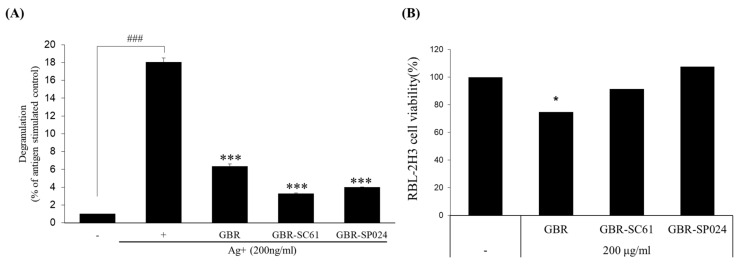
Effect of GBR-SC61 and GBR-SP024 on the degranulation of IgE/Ag-stimulated RBL-2H3 cells. − is untreated control, + is treated with anti-DNP IgE and stimulated with DNP-BSA. PP2 is an Src tyrosine kinase inhibitor. (**A**) Degranula-tion of RBL-2H3 cells was measured by the β-hexoaminidase assay (*** *p* < 0.001 vs. IgE/Ag-stimulated control and ^###^
*p* < 0.001 vs. nontreated control). (**B**) Cell viability of RBL-2H3 was measured by the Cell Counting Kit-8 assay. ANOVA was used to compare group means, followed by Dunnett’s *t*-test (* *p* < 0.05 vs. nontreated control). Data are expressed as means ± standard error of means (SEM) of three independent experiments (*n* ≥ 3).

**Figure 3 microorganisms-09-01855-f003:**
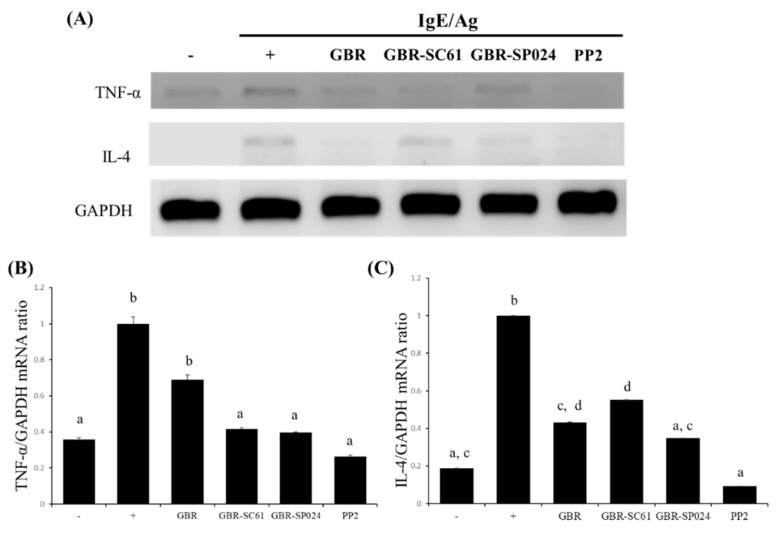
Effect of GBR, GBR-SC61, and GBR-SP024 on IgE/Ag-induced pro-inflammatory cytokine mRNA levels. (**A**) TNF-α and IL-4 mRNA expression was determined by reverse transcription-polymerase chain reaction (RT-PCR). GAPDH was used as the internal control. (**B**,**C**) TNF-α and IL-4 mRNA expression was normalized to GAPDH mRNA expression in each sample. Data are expressed as means ± standard error of means (SEM) of three independent experiments (n ≥ 3). Images were analyzed by one-way ANOVA and Duncan’s *t*-test (*p* < 0.01). Different letters indicate significant differences between groups.

**Figure 4 microorganisms-09-01855-f004:**
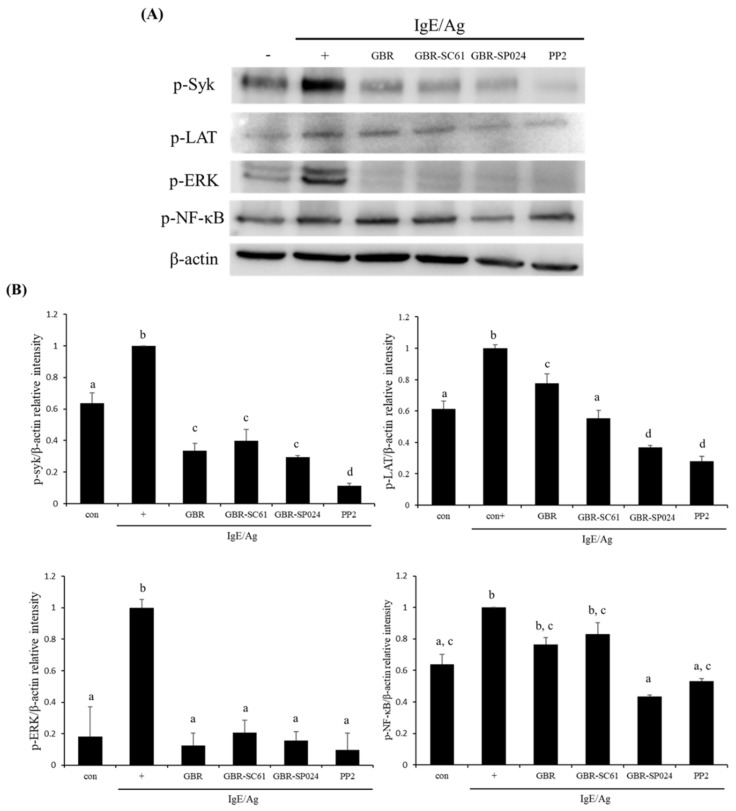
Effect of GBR-SP024 on the levels of phosphorylated proteins in IgE/Ag-stimulated RBL-2H3 cells. The levels of p-Syk, p-LAT, p-ERK, and p-NF-κB proteins were measured by immunoblotting. PP2 is a general Src family kinase inhibitor. (**A**) Western blot assay of p-Syk, p-LAT, p-ERK and p-NF-κB. (**B**) Expression level of GBR, GBR-SC61 and PP2. The levels were normalized to β-actin level. Representative images from three independent experiments are shown. Data are expressed as means ± standard error of means (SEM). Images were analyzed by one-way ANOVA and Duncan’s *t*-test (*p* < 0.05). Different letters indicate significant differences between groups.

**Figure 5 microorganisms-09-01855-f005:**
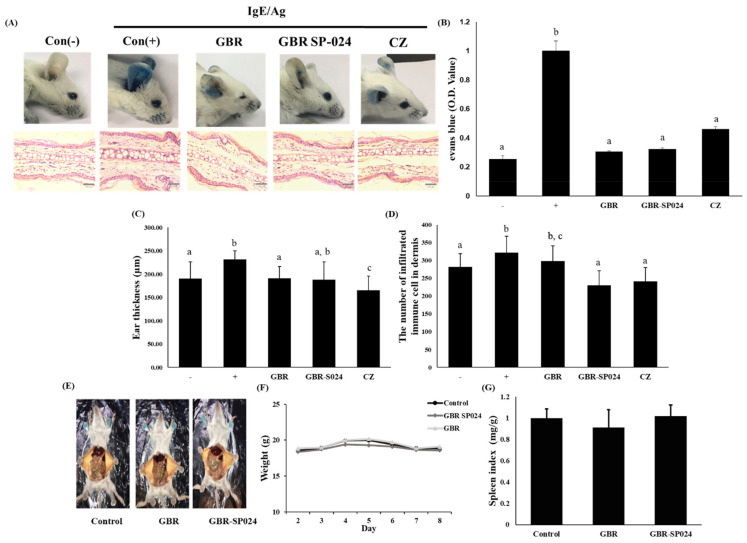
Inhibitory effect of GBR and GBR-SP024 on the IgE/Ag-mediated PCA reaction in BALB/c mice. GBR, GBR-SC61, and GBR-SP024 were orally administered to mice (200 mg/kg) after sensitization with DNP-IgE (0.5 μg). H&E staining was performed on BALB/c mouse ears in IgE/Ag-mediated PCA. Body and spleen weight changes in BALB/c mice after oral administration of GBR and GBR-SP024 were measured. Control (−) is untreated control, control (+) is treated with IgE and antigen, and cetirizine (CZ) is an established drug for type I hypersensitivity treatment. (**A**) Representative pictures of ears are shown after the PCA reaction. (**B**) Amount of extravasated dye from mouse ears. (**C**) Ear thickness measurement. (**D**) The number of infiltrated immune cells in dermis. (**E**) Mice were analyzed macroscopically after sacrifice on day 7. Representative pictures of mice are shown. (**F**) Changes in body weights. (**G**) Changes in spleen weights. One-way ANOVA was used to compare group means, followed by Duncan’s *t*-test (*p* < 0.01). Different letters indicate significant differences between groups. Data are expressed as means ± standard error of means (SEM) from total 6 mice/group representative of three independent experiments.

**Figure 6 microorganisms-09-01855-f006:**
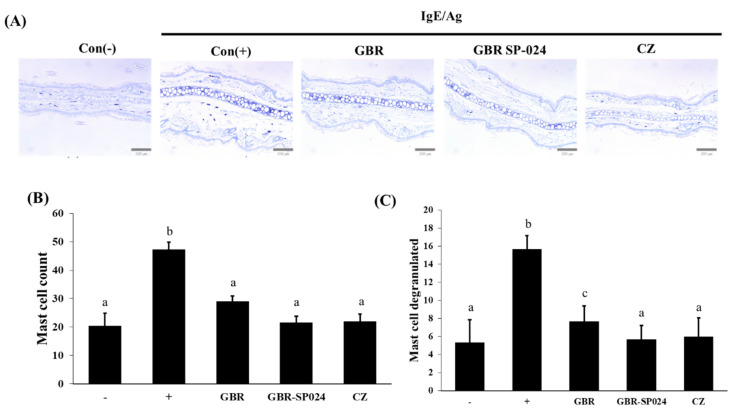
Histological examination of BALB/c mouse ears with toluidine blue stained on IgE/Ag-mediated PCA reaction in BALB/c. (**A**) Inflammatory edema image of ear tissue (**B**) The number of degranulated mast cells and (**C**) The number of mast cells in ear tissue. ANOVA was used to compare group means, followed by Duncan’s *t*-test (*p* < 0.01). Different letters indicate significant differences between groups. Data are expressed as means ± standard error of means (SEM) from total 6 mice/group representative of three independent experiments.

**Figure 7 microorganisms-09-01855-f007:**
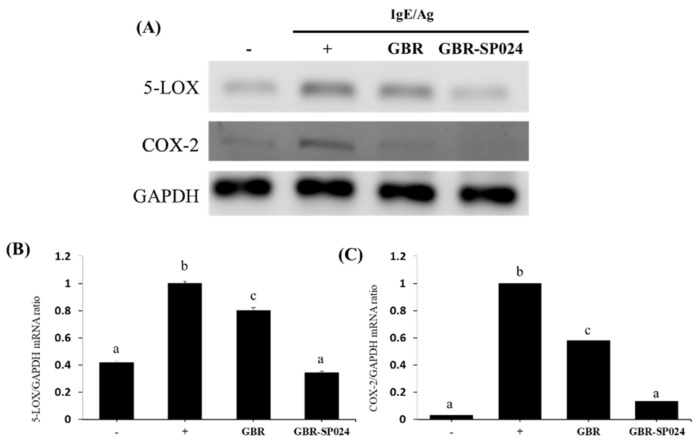
Effect of GBR and GBR-SP024 on 5-LOX and COX-2 mRNA expression in IgE/Ag-mediated PCA in BALB/c mouse ear tissues. (**A**) mRNA expression of 5-LOX and COX-2 was determined by RT-PCR. GAPDH was used as the control. (**B**,**C**) The results of the densitometric analyses of gel images are shown. The ratio of the mRNA densitometric signal relative to that of the internal control (GAPDH) was calculated. One-way ANOVA was used to compare group means, followed by Duncan’s *t*-test (*p* < 0.05). Different letters indicate significant differences between groups. Data are expressed as means ± standard error of means (SEM) from total 6 mice/group representative of three independent experiments.

**Figure 8 microorganisms-09-01855-f008:**
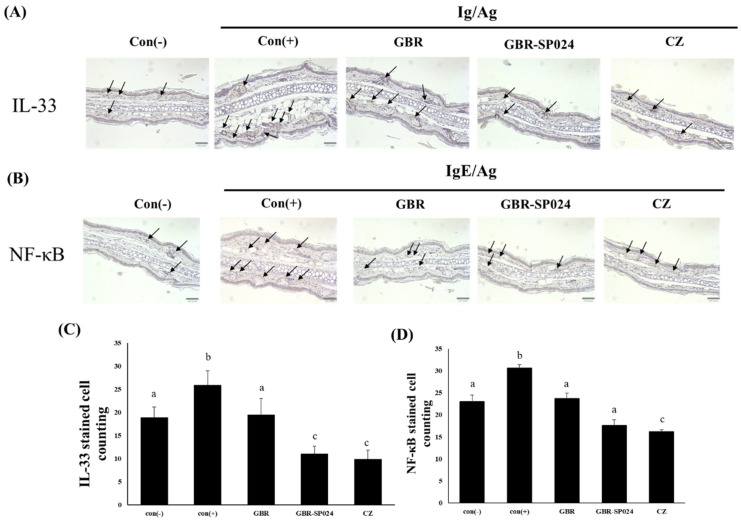
Immunohistochemical staining for IL-33 and NF-кB in BALB/c murine ears in IgE/Ag-mediated PCA. (**A**,**B**) Representative images of immunohistochemical staining for IL-33 and NF-κB expression (black arrow: stained cell). (**C**,**D**) IHC stained cell counting of IL-33 and NF-κB. ANOVA was used to compare group means, followed by Duncan’s *t*-test (*p* < 0.01). Different letters indicate significant differences between groups. Data are expressed as means ± standard error of means (SEM) from total 6 mice/group representative of three independent experiments.

**Table 1 microorganisms-09-01855-t001:** DPPH radical scavenging activity of GBR fermented with different lactic acid bacteria strains. Ascorbic acid was used as positive control. One-way ANOVA was used to compare group means, followed by Dunnett’s *t*-test (^$$^
*p* < 0.01, ^$$$^ *p* < 0.001 vs. ascorbic acid) Data are expressed as means ± standard error of means (SEM) from three independent experiments (*n* ≥ 3).

No	Strain	DPPH Activity (%)
1	Leuconostoc lactis S-Per.s12	64.0 ± 2.9 ^$$$^
2	Pediococcus pentosaceus SC7	78.0 ± 1.5
3	Lactobacillus paraplantarum SC61	67.6 ± 3.2 ^$$$^
4	Weisella kimchii Bro14	73.9 ± 13.3
5	Leuconostoc lactis S.Pum21	62.4 ± 4.0 ^$$^
6	Pediococcus pentosaceus ON-30A	53.8 ± 2.3 ^$$$^
7	Pediococcus pentosaceus GO008	82.0 ± 2.8 ^$$$^
8	Pediococcus pentosaceus SP-024	86.7 ± 2.7
	**Positive control**	**DPPH activity (%)**
	Ascorbic acid	81.1 ± 2.1

**Table 2 microorganisms-09-01855-t002:** Quantitative analysis of the trans-ferulic acid and protocatechuic acid content in GBR and GBR-SP024. One-way ANOVA was used to compare group means, followed by Dunnett’s *t*-test (* *p* < 0.05, ** *p* < 0.01, *** *p* < 0.001 vs. GBR). Data are expressed as means ± standard error of means (SEM) from three independent experiments (*n* ≥ 3).

Compound	Liquid Culture		Solid Culture	
	GBR	GBR-SP024	GBR	GBR-SP024
protocatechuic acid (mg/100 g)	0.1 ± 0.00	0.3 ± 0.0 **	not detecable	not detecable
*trans*-ferulic acid (mg/100 g)	23.2 ± 0.7	28.75 ± 0.8 ***	20.4 ± 0.7	22.9 ± 0.2 *

Data are expressed as mean ± standard error of means (SEM).

## Data Availability

Not applicable.

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
