# Peer review of "Pediococcus Pentosaceus from the Sweet Potato Fermented Ger-Minated Brown Rice Can Inhibit Type I Hypersensitivity in RBL-2H3 Cell and BALB/c Mice Models"

_microorganisms, 2021, doi:10.3390/microorganisms9091855_

Round 1

Reviewer 1 Report

The authors have answered to all questions raised  and made adequate changes to the manuscript, hence I recommend revised  version of the manuscript for publication in your journal. 

Author Response

To reviewer

Thank you

Reviewer 2 Report

Overall I find the study interesting. The findings might lead to the development of new treatment options for type 1 hypersensitivity. I agree that further studies are necessary especially studies using fractioned extracts. The anti-inflammatory effects can be caused by the synergistic influence of many compounds found in the extract.

I do have some questions/remarks:

  • I wonder if there was no data variation between tested samples? Some results have SEM of 0.0 which is rarely seen especially considering that the results are significantly different than the control. 
  • The last line of the conclusion "These find-507 ings suggest that GBR-SP024 is a potential target for the treatment of IgE-dependent type 508 I allergic diseases." should be rephrased since the GBR-SP024 is extract rather than a target.
  • In PCA study were the ears measure somehow or the whole cut ear was extracted?

Author Response

To reviewer

Thank you

This manuscript is a resubmission of an earlier submission. The following is a list of the peer review reports and author responses from that submission.

Round 1

Reviewer 1 Report

The authors presented results of a very interesting study investigating the effects of functional food (fermented germinated brown rice) on the IgE-mediated (type I) allergic reaction. While the topic of the study is of growing interest for the food industry and professionals involved in planning healthy and balanced diets for broader populations, and the study design was well planned and appropriate for the scientific questions raised, there are some major issues in reporting results (statistics) and conclusions drawn from those results. Authors should provide additional statistical evidence for the grater PCA attenuating effect of GBR-SC61 then the one observed for GBR, or alternatively adjust the conclusions raised in this manuscript.

Major issues:

  1. The authors did not report if they have assessed normality of data distribution, and how they have chosen parametric test (one-way ANOVA)

  1. In the material and methods part of the manuscript they stated “Results are shown as means ± standard errors of the means (SEM)”, while across the manuscript they report mean±Standard error (i.e. Table 2) and mean±Standard deviation (Figure 1-4)

  1. The authors should report final number of samples/animals (N) used for the statistical analysis and drawing of bar-graps presented in the figures.

  1. line 256-258 – Authors reported “GBR-SP024 significantly suppressed the levels of TNF-α and IL-4 mRNAs in 255 IgE/Ag-treated cells, compared with GBR and GBR-SC61 (p<0.05, p<0.001 vs. IgE/Ag-stimulated control). Therefore, we chose GBR-SP024 as the primary LAB strain for further experimentation.”

The authors have no statistical evidence to make such conclusion (all differences found by Dunnett’s t-test were statistically significant compared to IgE/Ag-stimulated controls)

  1. Line 274-276 – The authors reported “Indeed, GBR-SP024 suppressed the levels of phosphorylated Syk, LAT, ERK, and NF-κB proteins, compared to GBR and IgE/Ag-stimulated controls (Fig. 4).”

The authors have no statistical evidence to make such conclusion (all differences found by Dunnett’s t-test were statistically significant compared to IgE/Ag-stimulated controls).

  1. Line 293-295 – GBR-SP024 treatment (200 mg/kg) significantly suppressed extravasation of Evans blue dye in the IgE/Ag-stimulated ear, compared to GBR treatment (p<0.05, PCA, Fig. 5A and B).><0.05, PCA, Fig. 5A and B)”

The authors have no statistical evidence to make such conclusion (all differences found by Dunnett’s t-test were statistically significant compared to PCA or non-treated controls). Even by looking on the presented bargraphs, the effects of GBR and GBR-SP024 are similar.

  1. Line 304-307 “GBR-SP024 treatment was more effective than GBR treatment for decreasing the number of infiltrated inflammatory cells (72.0%±7.0 vs. 92.0%±8, p<0.001 vs. PCA).><0.001 vs. PCA).

The authors may say that the decrease in number vas greater in GBR-SP024 than in the GBR treatment but they cannot use the p values for the differences found between GBR-SP024 or GBR and PCA control).

  1. Line Furthermore, GBR-SP024 treatment significantly decreased the number of degranulated mast cells (45.8%±2.5), compared to GBR treatment (61.3%±7.4, p<0.001 vs. PCA).><0.001 vs PCA).”

Same comment as for the points 4-7.

  1. Figure 6 legend is missing.

  1. Line 347-349 “Furthermore, the mRNA levels of 5-LOX in IgE/Ag-stimulated mice treated with GBR-SP024 (63.8%±3.2) were significantly decreased compared to those in mice treated with GBR (20.1%±3.6, p<0.01 vs. PCA).><0.01 vs PCA).”

Same comment as for the points 4-8.

  1. Line 365-366 – “GBR-SP024 treatment also significantly suppressed IL-33 protein expression in 365 IgE/Ag-stimulated mice, compared with GBR treatment (p<0.001 vs. PCA).><0.001 vs. PCA).”

Same comment as for the points 4-8 and 10.

Minor issues:

  1. The authors should explain positive and negative controls used in their experiments, as well as why they used PP2 (i.e. Figure 2, 3…)
  2. Figure 4 - Were the WB data reported here normalized to β-actin? If yes, this should be mentioned in figure description. Authors are encouraged to replace figures with bargraphs as Figure 4, panel B. Definition of ** and * is missing in the figure legend.
  3. Figure 5. - ### is not defined in the figure legend. Please define controls (+, -, CZ??). The authors use PCA and IgE/Ag-stimulated control inconsistently.
  4. Line 321-322 – “We checked whether GBR- 321 SP024 could attenuate the infiltration and mast cells in the ear tissues (Fig. 4A).”

Is and correct, or it should be of? Is Fig. 4A accurate?

Reviewer 2 Report

In this manuscript, Dhong et al found the anti-allergic effect of GBR fermented with the SP024. In IgE/Ag-stimulated RBL-2H3 cells, the release of b-hexosaminidase and mRNA expressions of TNFa and IL4 are inhibited by GBR-SP024. In in vivo PCA mouse model, GBR-SP024 treatment reduced the infiltration of mast cells into dermis. In parellel, COX-2, 5-LOX and IL-33 expressions are reduced by GBR-SP024 administration. This is an interesting study to explore the usage of fermented GBR as dietary supplement for treating IgE dependent allergic disease. However, several major questions remain to be addressed.

  1. In general, the GBR-SP024 is not superior to GBR in antagonizing IgE dependent allergy, especially in in vivo study.
  2. The screening of SC61 and SP024 as the best LAB strains for fermenting BGR is not fully supported by the data. In table1, SC61 is not a outstanding one, why authors choose it rather than GO008 and SC7. Please show the result of 55 strains that have been tested, some of them can go to supplement.
  3. What’s the rationale to use as high as 200ug/ml in in vitro culture? Is there any titration experiment to test different doses?
  4. Please correct the errors in figure legends and add the Figure6 legend. Please correlate the results with figures correctly. The existing numerous errors really make it difficult to read and understand the logic of authors.
  5. please consider to use dots instead of bar in all the graphs. I am asking because the number of samples per group per experiment is not clear. 
  6. One additional experiment to sort Mast cell from ear and to run RNAseq will help a lot to address the mechanisms. The current results, including IL-33, cox-2 and 5-lox are very superficial.